# The contributions of entorhinal cortex and hippocampus to error driven learning

Shih-pi Ku [1,2✉], Eric L. Hargreaves[3], Sylvia Wirth [4] & Wendy A. Suzuki [1✉]

Computational models proposed that the medial temporal lobe (MTL) contributes importantly to error-driven learning, though little direct in-vivo evidence for this hypothesis exists. To test this, we recorded in the entorhinal cortex (EC) and hippocampus (HPC) as macaques performed an associative learning task using an error-driven learning strategy, defined as better performance after error relative to correct trials. Error-detection signals were more prominent in the EC relative to HPC. Early in learning hippocampal but not EC neurons signaled error-driven learning by increasing their population stimulus-selectivity following error trials. This same pattern was not seen in another task where error-driven learning was not used. After learning, different populations of cells in both the EC and HPC signaled long-term memory of newly learned associations with enhanced stimulus-selective responses. These results suggest prominent but differential contributions of EC and HPC to learning from errors and a particularly important role of the EC in error-detection.

[1] Center for Neural Science, New York University, New York, NY, USA. [2] Leibniz Institute for Neurobiology, Magdeburg, Germany. [3] Division of Neurosurgery, Rutgers University -- Robert Wood Johnson Medical School, New Brunswick, NJ, USA. [4] Institut des Sciences Cognitives Marc Jeannerod, UMR 5229, Bron Cedex, France. ✉email: shihpi@gmail.com; ws21@nyu.edu

A large body of behavioral work supports the idea that feedback from errors is an important teaching signal to help supervise learning, adjust behavior and achieve learning goals[1–4]. Generally, errors are defined as the difference between the received and expected outcome after the execution of a particular behavior. Seminal work by Rabbitt and colleagues in the mid-1960s[5–8] first raised the importance of a system that detected errors to adjust performance. Evidence for neural signals that processed errors first appeared in the early 1990s with the observation of negative electrical potential that occurred in the medial frontal region of the brain, about 50 to 100 ms after making an error, termed error-related negativity[9,10]. Combining error-related negativity measurement and functional magnetic resonance imaging (fMRI), a prominent error-detection network has been described in humans[11,12]. Areas including anterior cingulate cortex (ACC), anterior insular (operculum), ventral lateral prefrontal cortex, dorsal lateral prefrontal cortex and parietal lobe have been reported to contain signals sensitive to errors, with the ACC most consistently found to be involved in error detection. Using behavioral neurophysiology, various groups have also identified prominent error-detection signals in the ACC of non-human primates[10,11,13–20], consistent with human literature.

Recent work suggests that this error-detection network may extend to structures within the medial temporal lobe (MTL). Parallel findings in humans, monkeys, and rats suggest that structures of MTL not only participate in encoding both error and correct signals (termed outcome-selective cells) but that a subset of these outcome-selective cells also signal learning as well. Using an object-place associative learning task (OPT), Wirth et al.[21] reported that half of the neurons recorded in the monkey hippocampus differentiated between correct and error trials during the inter-trial interval period of the task. While about half of the outcome-selective hippocampal neurons responded preferentially to erroneous outcome (error-up cells) the other half responded preferentially to correct outcome (correct-up cells). Further analysis showed that while the correct-up cells also increased their stimulus-selective response after learning, the error-up cells did not, suggesting that the correct-up but not error-up cells also participate in the learning process. Another study reported prominent trial outcome signals in both the entorhinal cortex (EC) and hippocampus (HPC) in monkeys and humans performing the same associative learning task using local field potentials and blood-oxygen-level-dependent (BOLD) fMRI approaches[22], respectively. In that study, however, the relationship between outcome signals and learning was not examined. Similarly in rodents, Ahn and Lee[23] reported error-related activity in another medial temporal area, the perirhinal cortex, during the performance of an object-target association task. In contrast to the Wirth et al.[21] finding, this group showed that error-up outcome cells carried more information about the learned associations immediately after error trials than after correct trials.

Work by Lorincz and Buzsaki[24] and Ketz et al.[25] suggest a computational framework with which to understand these reports of error detection and learning signals throughout the MTL. Their models outline an anatomically specific three-step process underlying error-driven learning. It starts with error detection in the EC. The EC error signals then induce synaptic modification in areas CA3 and CA1 of HPC to correct for the errors as the second step. Third, that modified hippocampal synaptic output is proposed to entrain long-term memory traces in the EC. The idea that a memory trace is first rapidly formed in HPC then transferred to EC is consistent with system consolidation theory[26–30]. The goal of the present study is to examine the role of the EC and HPC in error-driven learning and the early consolidation process

and to test the predictions of the models of Lorincz, Buszaki[24], and Ketz[25]. We examined the behavioral and neurophysiological responses in these areas as monkeys performed an associative learning task in which they used an error-driven learning strategy (i.e., they performed significantly better after an error trial relative to after a correct trial). We report that during the task in which the animals used an error-driven learning strategy, we observed both error-detection signals as well as associative learning signals in both HPC and EC that are consistent with some, but not all of the predictions from Lorincz, Buszaki, and Ketz[24,25]. By contrast, when we examined neural activity in HPC from a different associative learning task where animals used a correct-based learning strategy (i.e., animals perform significantly better after a correct trial than after an error trial), we did not find the same prominent error-based learning signal.

## Results

**An error-driven learning strategy was used in the location-scene task.** To test whether and how MTL was involved in error-driven learning, we recorded HPC and EC in three and two monkeys, respectively while they performed a conditional motor associative learning task (location-scene associative learning task, or LST; Fig. 1a). In this task, each day, monkeys learned to associate a visual cue to a particular rewarded target location through trial-and-error (Fig. 1b). Across 373 sessions, the five monkeys saw 1556 new scenes, of which they learned 1015. They learned in average 3.1 (SD = 0.11) location-scene associations per session and needed on average 10.6 (SD = 0.55) trials to learn each new association to criteria.

To determine whether animals used an error-driven learning strategy to perform the LST, we asked whether the monkeys performed better on the trials immediately following errors compared to the trials immediately following correct responses (Fig. 1c), excluding all trials the animals aborted or did not make a target selection[31]. Because there were almost always fewer error trials than correct trials in one session (99% of cases), and most of the errors occurred during the early acquisition stage of learning, we used the same number of correct trials as error trials from the beginning of the session to calculate the mean behavioral performance immediately after either correct or error trials. We termed this period of learning the "memory acquisition stage" (see below). We included 282 behavioral data sets from five monkeys where they learned at least one location-scene association and made at least 20 error trials in one session. Across all sessions, the averaged performance immediately after error trials was 74.46% (SD = 12.26%), which was significantly better (paired $t$-test, $t(281) = 4.62$, $p < 0.001$, $d = 0.39$) than the performance seen after correct trials ($M = 69.76\%$, SD = 11.77%, Fig. 1c, individual data points are shown in Supplementary Fig. 7a), suggesting that monkeys used an error-driven learning strategy to perform this task.

**Error-detection signals in EC and HPC.** Since the animals used an error-driven learning strategy to perform LST, we asked whether we could see evidence for an error-detection signal in the neurophysiological responses of EC or hippocampal neurons in the same 5 monkeys used in the behavioral analysis. We previously showed that many hippocampal cells signaled trial outcome during the inter-trial interval of an object-place associative learning task[21] by either increasing their firing rate on correct trials relative to error trials (correct-up cells) or by increasing their firing on error trials relative to correct trials (error-up cells). To examine the correct-up and error-up signals in the HPC and EC during the performance of the LST, we analyzed 135 hippocampal neurons and 143 entorhinal neurons (recording sites see

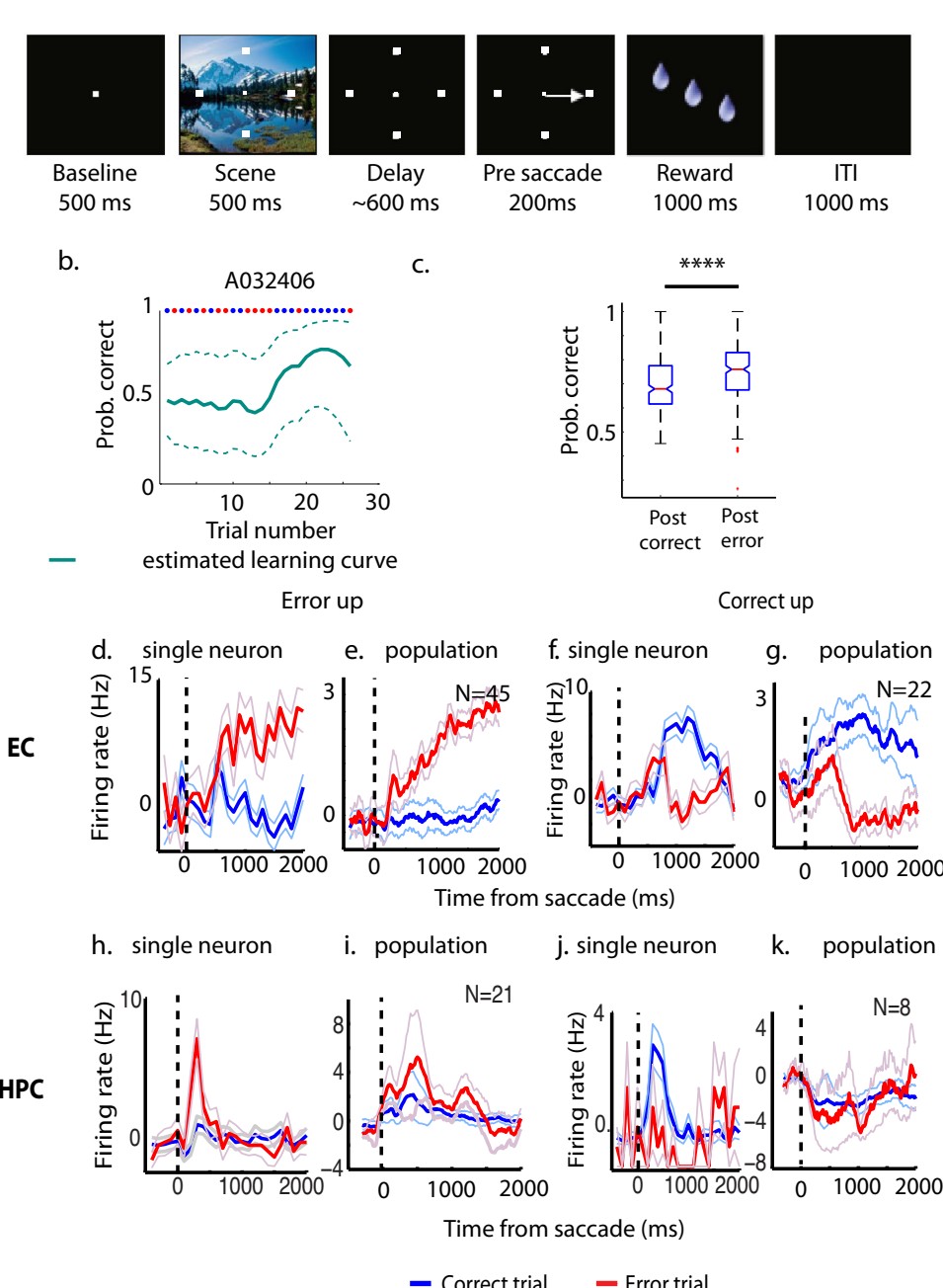

**Fig. 1 Location-scene task (LST), performance, and outcome-selective signals. a** Schematic illustration of the LST. **b** Estimated performance of one example LST session. The red dots at the top of the graph indicate error trials and the blue dots indicate correct trials. The teal dashed lines indicated the 95% confidence intervals of the estimate. Prob.: probability **c** Averaged performance immediately after error trials is significantly better than after correct trials during LST (averaging across 282 sessions, 5 monkeys. Central red marks indicate the median, and the bottom and top edges of the boxes indicate the 25th and 75th percentiles, respectively. The whiskers extend to the most extreme data points not considered outliers, and the outliers are plotted individually using the "+" symbol. Notches of the boxes indicate the 95% confidence intervals of the estimated median. Individual data points are shown in Supplementary Fig. 7a and data points are listed in Supplementary Data 1). **d** Time courses of one representative error-up cell in EC. The time courses are averaged across either error trials (red) or correct trials (blue). **e** Averaged and normalized time courses of population EC error-up cells. **f** Time courses of one representative correct-up cell in EC. **g** Averaged and normalized time courses of population EC correct-up cells. **h** Time courses of one representative error-up cell in HPC. **i** Averaged and normalized time courses of population HPC error-up cells. **j** Time courses of one representative correct-up cell in HPC. **k** Averaged and normalized time courses of population HPC correct-up cells. ****$p < 0.0001$. The light-blue and light-red lines indicate the SD of averaged time courses.

Supplementary Fig. 1). Similar to a previous publication[21], we defined outcome-selective cells based on their mean firing rate during the first and second 1000 ms of the inter-trial interval (ITI) separately. We determined whether the neurons responded significantly more to correct or error outcome during either of

these periods by performing a paired *t*-test. To correct for multiple comparisons, the significance threshold was set to $p < 0.025$. In EC, we found 46.85% (67/143) of the recorded cells were outcome-selective (i.e., neurons responded significantly differently to correct versus error outcome during the ITI). Of these,

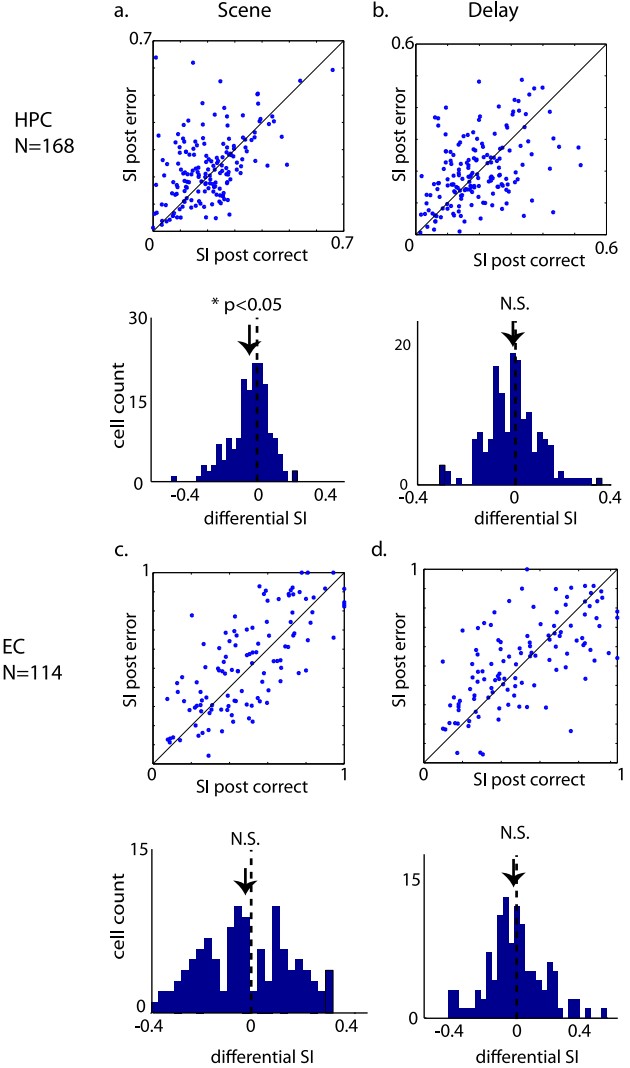

**Fig. 2 Illustration of selectivity index (SI) immediately after error trials versus after correct trials. a**, **b** The SI during scene and delay period in HPC. **c**, **d** The SI during scene and delay period in EC. The distribution of differential selectivity index (after correct—after error) is also plotted below each scatter plot. During the scene period in HPC the SI after error trials is significantly higher than after correct trials ($t(166) = 2.405$, $p = 0.0173$, paired $t$-test). All data points are shown in Supplementary Data 1.

67.16% (45/67 cells) signaled errors by responding to error outcomes with a significantly higher firing rate than correct outcomes, while the remaining 32.84% (22/67 cells) responded significantly higher to correct outcomes relative to error outcomes. By contrast, in the HPC, only 21.48% (29/135) of the hippocampal cells were outcome-selective with 72.41% (21/29 cells) signaling errors and the remaining 27.59% (8/29 cells) signaling correct trials. In both EC and HPC, the proportion of error-up cells was significantly higher than correct-up cells (chi-square test of two proportion difference, $X^2(1) = 10.31$, $p = 0.0013$ for EC, and $X^2(1) = 6.53$, $p = 0.01$ for HPC).

The representative time courses of one example error-up and one example correct-up cell in EC and HPC are shown in Fig. 1d, f, h, j, respectively. The averaged time courses of all error or all correct detecting cells in the EC or HPC are shown in Fig. 1e, g, i, k, respectively. To quantify the strength of error-up and correct-up cells to differentiate correct and error outcome, we used receiver–operating characteristic (ROC) analysis to estimate the information carried by each population in both the EC and HPC.

The averaged time courses and area under the ROC curve (AU-ROC) of all error-up (Supplementary Fig. 2a) and correct-up cells (Supplementary Fig. 2b) in EC showed that the outcome-selective signals in both groups were sustained, not transient signals. The differential response of the error-up cells was even maintained throughout the entire two second ITI included in the analysis. By contrast, the separation in hippocampal subtypes was relatively weak. The AU-ROC of error-up cells in HPC was significantly higher than 0.5 (Supplementary Fig. 2c, $t(22) = 2.26$, $p = 0.003$), however, it was significantly smaller than in EC ($t(63) = 6.46$, $p < 0.0001$, $d = 1.89$). The AU-ROC of correct-up cells in HPC was barely greater than 0.5 (Supplementary Fig. 2d, $t(7) = 1.05$, $p = 0.33$). The results show that outcome-selective signals (including error-detection signals) exist in both EC and HPC with a stronger overall representation in EC.

**Error-driven learning signals in EC and HPC.** Given the striking error-driven learning seen at the behavioral level, we next searched explicitly for error-driven learning signals in the neural activity of entorhinal and hippocampal cells. Given that behavior improves significantly more after error but not after correct trials during the memory acquisition stage (as defined previously, details in "Methods" section), we hypothesized that early in the learning process, cells in the EC or HPC might reflect this behavioral improvement with higher stimulus-selective visual responses during the scene or delay periods of the task immediately after error trials relative to correct trials. We analyzed a total of 114 entorhinal neurons from two monkeys and 168 hippocampal neurons from three monkeys recorded during the same 282 sessions used for the behavioral analysis (see "Methods" section for rational for the cell counts used). We found that during the memory acquisition stage (see definition above), hippocampal neurons showed significantly higher visual selectivity following error trials relative to correct trials during the scene period (paired $t$-test, $t(166) = 2.405$, $p = 0.0173$, Fig. 2a) but not in the delay period of the task (paired $t$-test, $t(166) = 0.918$, $p = 0.36$, Fig. 2b). In EC, the selectivity was not different following error trials relative to correct trials during either scene or delay periods (paired $t$-test, $t(126) = 0.969$, $p = 0.334$ for scene and $t(125) = 1.96$, $p = 0.052$ for delay, Fig. 2c, d). These results are consistent with predictions from computational models[24] that the error signals induce early plasticity in HPC.

To test the sensitivity of our selectivity analysis, we used a permutation test in which we permutated the labels 1000 times during the scene period of the task. Of the 168 hippocampal neurons, the SI of 102 out of 168 cells surpassed the 95% quantile threshold. Analyzing only these 102 neurons, the difference in the SI between post-error trials and the post-correct trials was clear ($t(101) = 2.79$, $p = 0.0062$, Supplementary Fig. 3a). Performing the same selection procedure to the delay period for hippocampal neurons, to EC neurons during either the scene or delay period yielded a differential SI that did not differ from zero (Supplementary Fig. 3b–d).

To test whether the shift of selectivity after error trials in the hippocampus was due to one single subject, we separately plotted the post-error vs. post-correct SI of individual subjects (Supplementary Fig. 4). We also performed the chi-square test of independence to see whether the variances of SI within subjects were consistent across subjects. The results were $X^2(166) = 45.2$, $p < 0.001$ for post-correct SI and $X^2(166) = 45.1$, $p < 0.001$ for post-error SI, indicating no-dependency of SI on different subjects and suggested that the increased SI after error compared to correct trials was consistent across subjects.

To test the specificity of this early hippocampal error-driven learning signal during the LST, we examined the behavior and

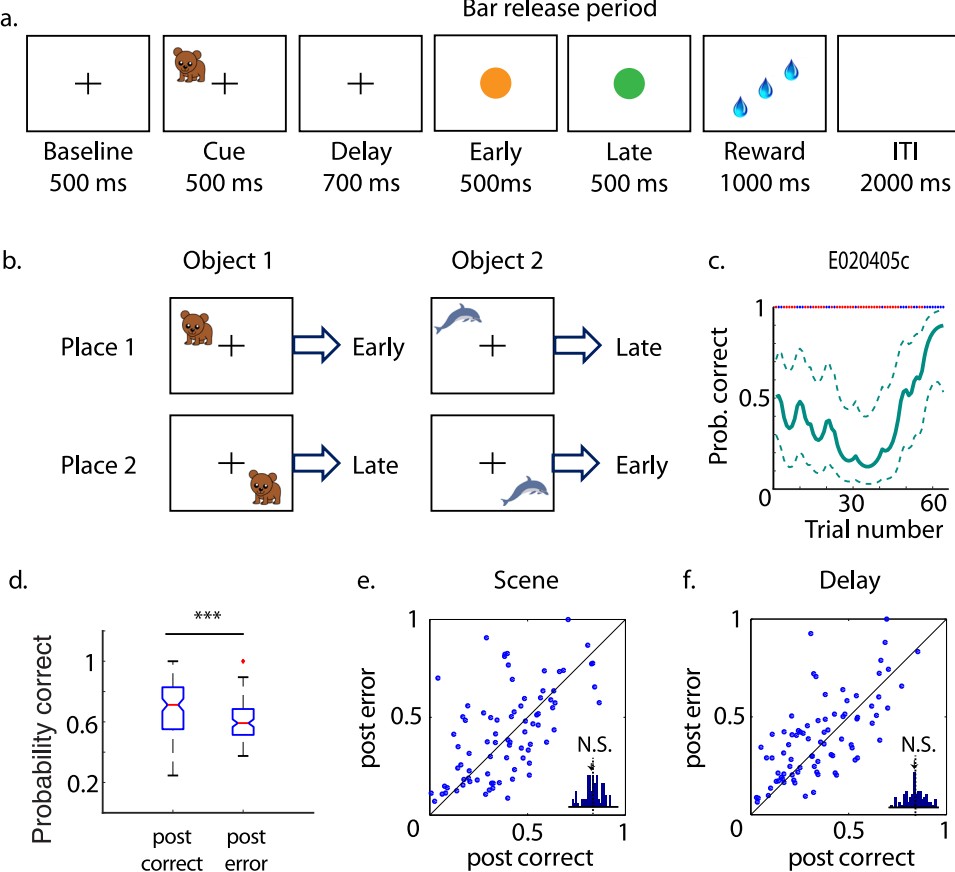

**Fig. 3 Object-place task (OPT), performance, and selectivity index (SI). a** Schematic illustration of the OPT. **b** Illustration of the object-place response contingencies used each day. **c** Estimated performance of one example OPT session. The red dots at the top of the graph indicate error trials and the blue dots indicate correct trials. The dashed lines indicate the 95% confidence intervals. **d** The averaged performance immediately after correct trials (0.7 ± 0.017) is significantly better than after error trials (0.61 ± 0.01) for the OPT (***$p < 0.001$, $t$-test. The labels of the boxplots are the same as in Fig. 1c. Individual data points are shown in Supplementary Fig. 7b). The SI immediately after error trials is not significantly different from after correct trials during either the scene (**e**) or delay (**f**) period of the task. All data points are shown in Supplementary Data 1.

neurophysiological responses in the HPC as two monkeys performed another associative learning task, the object-place task (OPT; Fig. 3a, b; one example learning curve in Fig. 3c, EC recordings were not done). This task has been described in detail before[21]. Briefly, monkeys learned to associate an object-place combination to one of two possible bar-release responses (early vs. late) to obtain rewards. Among the 133 sessions, the OPT was significantly more difficult than the LST task with animals learning significantly fewer object-place combinations per OPT session ($M = 1.99$, SD $= 0.1$) relative to the LST ($M = 3.1$, SD $= 0.11$, $t(299) = 90.48$, $p < 0.00001$, $d = 10.56$). Also consistent with this idea, animals took significantly more trials on average to learn each new associations in the OPT ($M = 16.08$, SD $= 1.34$ trials) than for the LST ($M = 10.57$, SD $= 0.55$ trials, $t(299) = 48.41$, $p < 0.00001$, $d = 5.38$).

We analyzed 86 behavioral data sets from two monkeys in which they made at least 20 error trials within each session and learned at least one object-place association (same criteria as for LST). Using the same trial sorting strategy as we used for the LST, we calculated the mean performance of trials immediately after error trials and after the same number of correct trials from the beginning of the learning session during the memory acquisition stage. The averaged performance immediately after correct trials across all sessions was 70% (SD $= 1.7$%), which was significantly better than that after error trials ($M = 61$%, SD $= 1$%, $t(170) = 42.32$, $p < 0.00001$, $d = 6.45$, Fig. 3d. Individual data points are

shown in Supplementary Fig. 7b). These findings suggested that, unlike the LST, these animals used a correct-trial-based strategy to learn the OPT. When we examined the selectivity of hippocampal cells after both correct and error trials, we found no increase in stimulus-selectivity after either correct or error trials (differential selectivity $M = 0.02$, SD $= 0.18$, $Z(170) = 0.53$, $p = 0.599$, paired $t$-test for scene and $M = 0.01$, SD $= 0.15$, $Z(170) = 0.43$, $p = 0.667$ for delay period; Fig. 3e, f). This suggests that the hippocampal shift in selectivity seen in the LST was specific to a task where an error-driven, but not a correct-driven learning strategy was used. Note EC cells were not recorded during the OPT task.

**Long-term associative learning signals in EC and HPC.** The third prediction of Lorincz and Buzsaki's[24] error-driven learning model indicates that the new learning signal generated by the HPC induced by entorhinal error inputs will train long-term memory traces in the EC. To test this hypothesis, we asked whether there were long-term memory signals seen in either the EC or HPC after learning occurred towards the end of the sessions[21]. We previously reported that during the performance of the OPT (in which animals used a correct-trial based learning strategy), correct-up cells signaled long-term memory for the learned object-place combinations by increasing their stimulus-selective visual response after learning relative to before learning,

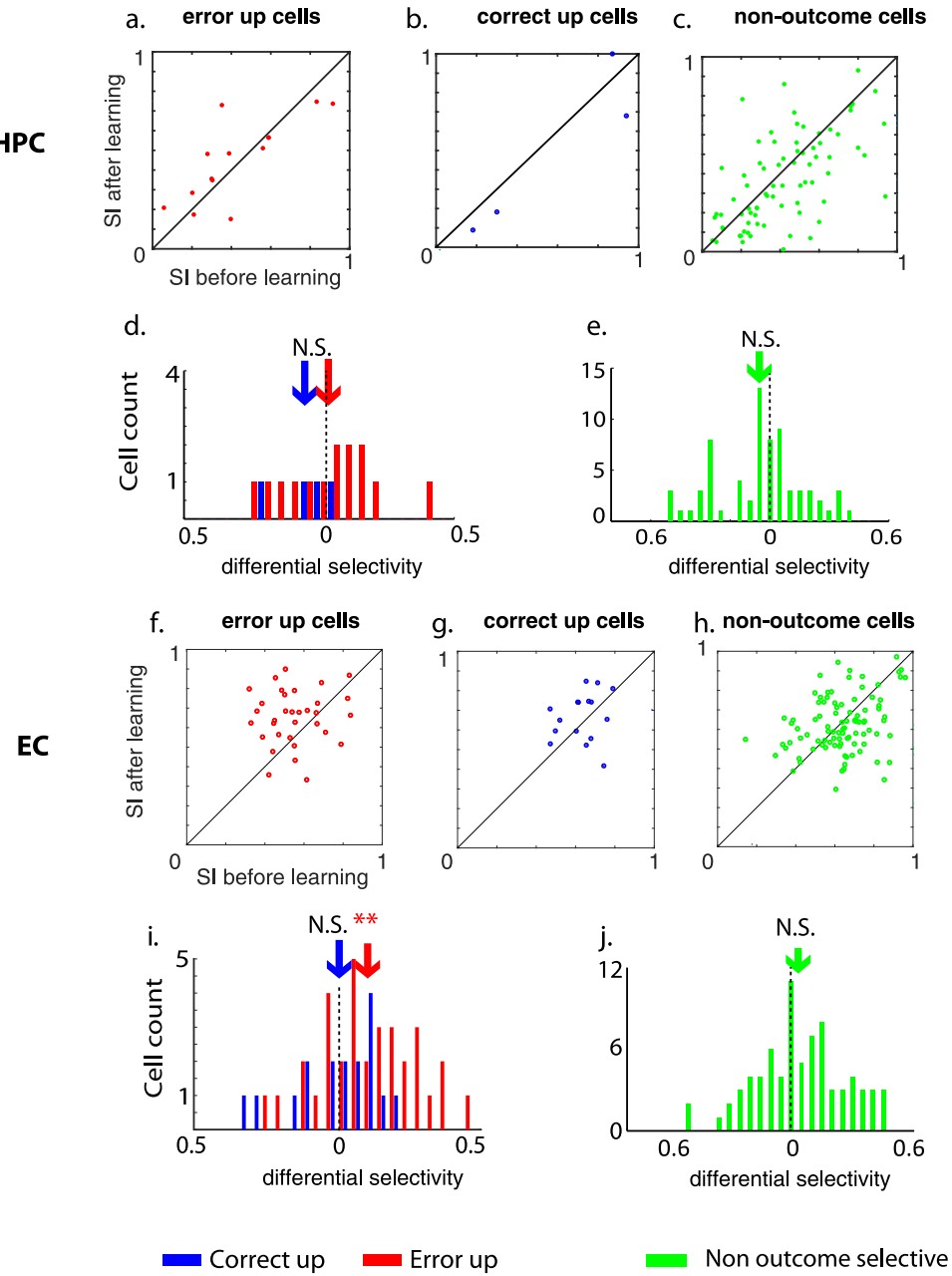

**Fig. 4 Error-up cells in EC increased their selectivity index (SI) after learning.** The SI during the delay period before versus after learning of the same error-up (**a**), correct-up (**b**), or non-outcome-selective neurons (**c**) in HPC are plotted against each other. The distribution of differential SI before and after learning for the error-up and correct-up cells (**d**) and non-outcome-selective cells (**e**) in HPC are also plotted to indicate no significant shift in the differential SI in HPC. The same information for cells in the EC is shown in **f–j**. In **i** the differential SI of error-up cells shows a significant right-ward shift. Arrows indicate the averaged differential selectivity of each population. **\*\***$p < 0.005$. The vertical dashed lines in **d**, **e**, **i**, and **f** indicate no differential selectivity. All data points are shown in Supplementary Data 1.

presumably signaling a more precise memory signal for the learned associations[21]. Based on these findings, we examined correct-up, error-up and non-outcome-selective cell populations in the EC and HPC during the LST for similar long-term memory signals. In HPC, neither the error-up, correct-up nor the non-outcome-selective cells changed their selectivity index with learning ($p > 0.05$, paired t-test, exact statistics see Supplementary

Table 1, Fig. 4f–j, Supplementary Fig. 6c, d). However, in the EC, we found that during the delay but not the scene period of the task, error-up cells increased their stimulus-selective response significantly after learning relative to before learning while the animals learned at least one scene-location association (33 sessions, $t(64) = 2.89$, $p = 0.0053$, $d = 0.71$, 2-sample t-test for the delay. Differential selectivity $M = 0.1$, SD $= 0.19$, $t(32) = 3.02$, $p$

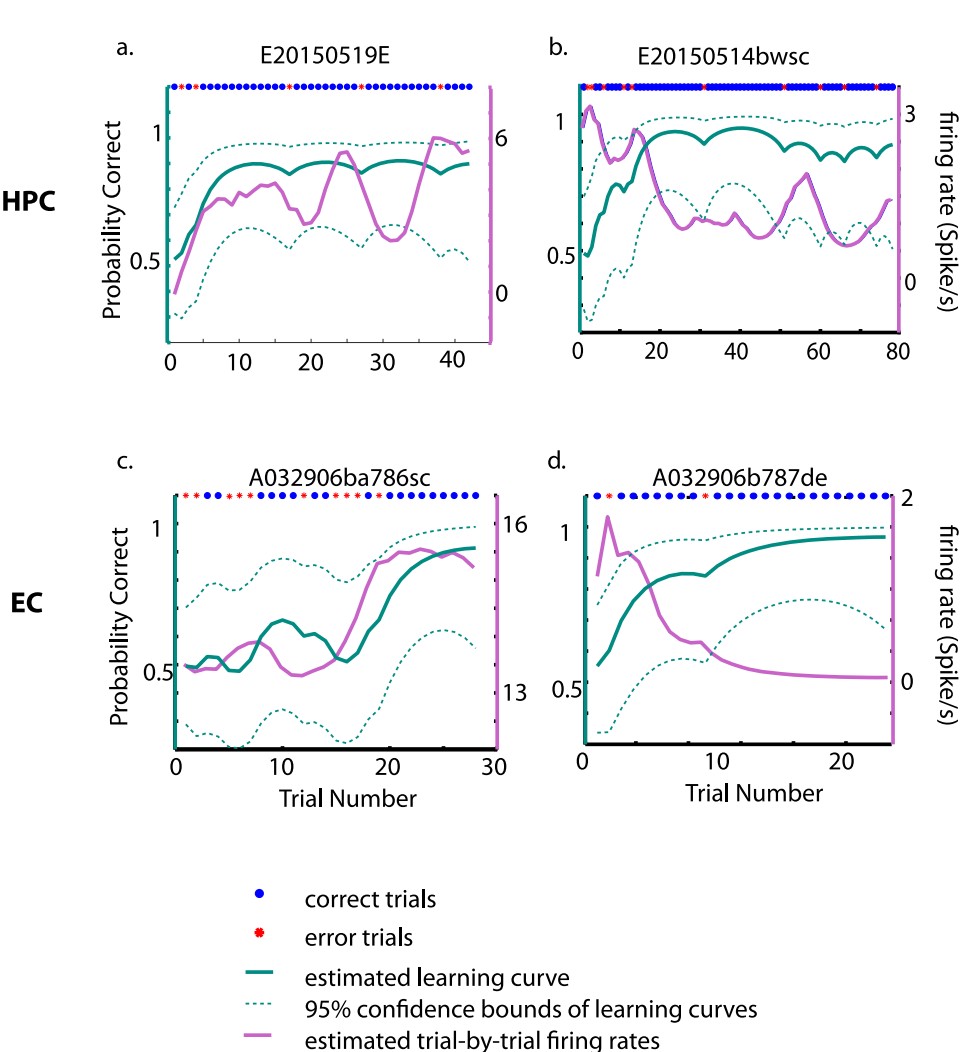

**Fig. 5 Estimated learning curves and firing rates of example changing cells.** The estimated trial-by-trial firing rates (purple lines) superimposed on estimated learning curves (teal lines) of an example sustained changing cell in HPC was plotted in **a**, a baseline sustained changing cell in **b**, an example sustained changing cell in EC in **c**, and a baseline sustained changing cell in **d**. The blue-filled circles on top of each subplot indicate correct trials and the red asterisks indicate the error trials. The 95% confidence bounds of estimated learning curves were plotted in teal dashed lines. The trial-by-trial firing rates plotted here were estimated with the same state-space algorithm used to estimate behavioral performance[33] to avoid the difficulty of recognizing the correlation between the firing rates and the behavior performances due to the noisy neural spiking with inherited variability. Performances are shown in Supplementary Data 1.

= 0.0049, paired *t*-test for delay (Fig. 4b, d). For scene: differential selectivity $M = 0.02$, SD = 0.18, $t(32) = 0.62$, $p = 0.534$, Supplementary Fig. 6a). To demonstrate the increase of selectivity after learning was not due to the neural activity of a single subject, we separately plotted the SI before and after learning of each individual subject (Supplementary Fig. 5). By contrast, neither the entorhinal correct-up cells (differential selectivity $M = 0.035$, SD = 0.167, $t(18) = 0.92$, $p = 0.371$, paired *t*-test for delay (Fig. 4a, d). For scene: differential selectivity $M = -0.084$, SD = 0.226, $t(18) = 1.62$, $p = 0.123$, Supplementary Fig. 6a) nor the entorhinal non-outcome-selective cells changed their selectivity indexes (differential selectivity $M = -0.032$, SD = 0.207, $t(98) = 1.51$, $p = 0.1334$, paired *t*-test for delay, Fig. 4c, e and for scene: differential selectivity $M = 0.0024$, SD = 0.21, $t(98) = 0.11$, $p = 0.909$, Supplementary Fig. 6b).

We also previously reported that cells in the HPC signaled long-term memory for learned location-scene associations by

changing their stimulus-selective activity correlated with learning the LST (changing cells[32]). Here we show that the same pattern of changing cells was seen in 17% of the newly recorded hippocampal cells used for this study (23/135 cells; Fig. 5a, b). This proportion is not different from the 17.24% of hippocampal changing cells previously reported in Wirth et al.[32] (25 changing cells from 145 recorded neurons, derived from Table 1 in Wirth et al., $X^2(1) = 0.0021$, $p = 0.96$, chi-square test of two proportion difference). Among the newly recorded hippocampal changing cells three are correct-up cells and three others are error-up cells. By contrast, only 8% (15/179, Fig. 5c, d) of entorhinal neurons were changing cells, which was significantly lower than in HPC ($X^2(1) = 5.65$, $p = 0.017$, chi-square test of two proportion difference). Among the entorhinal changing cells two are correct-up cells and three are error-up cells. Taken together, these results suggest that at the later stage of the learning sessions when performance is significantly above chance levels, neurons in both

the EC and HPC provide long-term representations of learned associative information.

## Discussion

Here we showed that neurons in the macaque EC and HPC play prominent but distinct roles in error-driven learning. First, we showed that in a task that used an error-driven learning strategy, error-detection signals (error-up cells) were observed in both the EC as well as HPC. However, we found significantly more error-detection cells and an overall stronger (i.e., more differential) error-detection signals in the EC relative to HPC. Second, we report early error-driven learning-related increases in stimulus-selective responses in the population of hippocampal cells but not the EC. This hippocampal shift in selectivity was specific to a task where animals used an error-driven learning strategy and were not seen in another associative learning task in which they used a correct-based learning strategy. Third, we show evidence for different types of long-term memory signals in the EC (enhanced selectivity in the error-up cells) and HPC (changing cells) after learning. We discuss each of these findings with respect to the time course of behavioral learning (Fig. 6a) and predictions from the computational models of error-driven learning in the MTL[24,25].

**Error detection in EC-HPC and its relation with other brain areas.** Perhaps the most surprising finding reported here is the prominent error-detection signals in EC (45/143 cells, 30% of recorded neurons in EC), with a smaller proportion of EC neurons signaling correct outcome (correct-up cells, 22/143 cells, 15%). These prominent EC error-up signals support the first prediction of computational models of error-driven learning signals[24] and suggest that EC is part of the error-detection network identified in the human and non-human primate brains[11,13,34] with the best-studied error-related activity described in ACC[13,35]. In monkeys, the early studies showing error-detection signals did not use learning tasks but still identified error-detection signals in ACC. Using a saccade-countermanding task, a substantial proportion of neurons recorded in ACC showed selective and sustained activity after the animals made an erroneous saccade[17]. During a voluntary movement selection task, a subset of rostral cingulate motor area cells (part of ACC) fired spikes persistently over several hundred milliseconds for decreased reward relative to the previous trials[36]. Recently a study recording ACC and lateral habenula in monkeys performing a reversal-learning task reported about half of the neurons in ACC encoded trial outcomes, and nearly 80% of lateral habenula neurons preferred negative relative to positive outcome[37]. In this study, the animals needed to use the trial outcome history in order to know which saccade target was associated with higher reward probability in the current trial. Similarly, in the present study, the monkeys needed to estimate which saccade target was associated with a specific visual stimulus by learning the reward contingency through a trial-and-error procedure. The error-detection neurons in lateral habenula in Kawai's study show increased responses with no reward trials and the same neurons decreased their responses with reward trials. In contrast, and similar to the pattern of activity seen in the EC in the present study, ACC neurons tended to fire preferentially to either positive or negative outcomes and did not decrease their firing rate during opposite outcome periods. Timing differences between ACC and EC, however, have been reported. For example, several previous studies in ACC reported that the error outcome signals seemed to rise within 200 ms after an error response[16,38]. By contrast, we report that in EC cells, a clear separation between an error from correct outcome started only about 500 ms after the erroneous

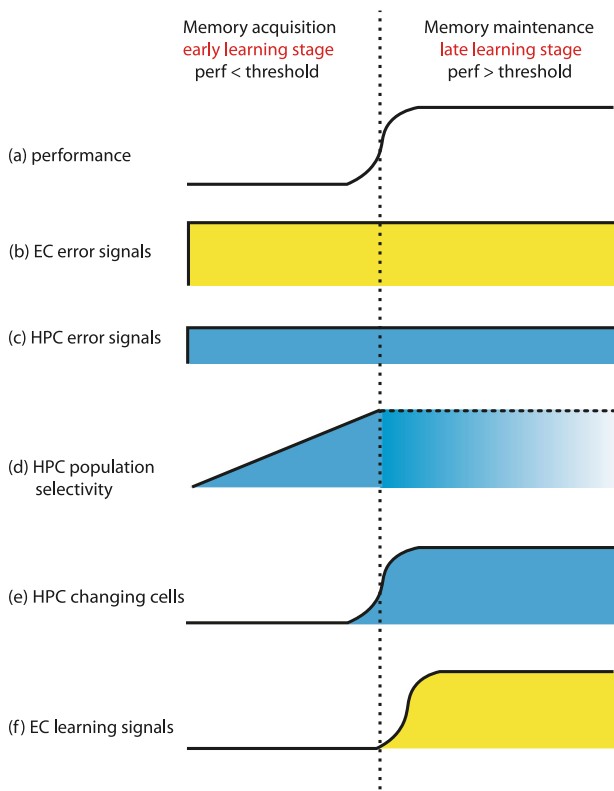

**Fig. 6 Schematic diagram and time courses of error-detection and learning signals in the EC and HPC.** Based on the performances of the animals (**a**), learning could be divided into early and late stages: During the early stage of learning (left panel), also termed the memory acquisition stage in the present study, the performance has not reached the learning criteria and the animals still made a lot of behavioral errors. Error signals in EC (**b**, yellow) and HPC (**c**, blue) are prominent and do not change over the course of learning, though the error singles are stronger in EC relative to the HPC. The population selectivity in HPC increased after error trials readily in this early stage (**d**) and cannot be determined during the later stage of learning because there are not enough error trials (usually fewer than 10 errors). Late in learning (right panel), also termed the memory maintenance stage, the performance surpasses the threshold and the error signals in both EC and HPC maintained high (**b**, **c**). The learning signals in HPC develop into a sparser representation as changing cells (**e**) while in EC learning signals also emerged in the error-up cells, which are at the same time the error-detection cells (**f**). perf: performance.

choice of saccade direction (Supplementary Fig. 2a) suggesting that the EC outcome signal may originate in the ACC. The anatomical connections between the ACC and EC are strongly bidirectional[39–42] with ACC projections terminating in the mid-to deep layers of EC. Previous studies suggest that that ACC might play a role in mediating mnemonic functions in MTL by gating the input-output efficiency between EC and perirhinal cortex[43]. Further studies involving simultaneous recording of the ACC and EC will be of great interest to determine how these regions might interact with each other to regulate the mnemonic or further cognitive functions.

There is also a detectable but decidedly smaller error signal in HPC relative to the EC (Supplementary Fig. 2 and Fig. 6b vs. Fig. 6c). Unlike the strongly bidirectional projections between the ACC and the EC, the projections between the HPC and ACC are largely unidirectional with HPC neurons projecting to the ACC[44]. Similar to the present study, Wirth et al. also reported error-

detection signals in HPC while the monkeys performed the object-place associative learning task (OPT, Fig. 3[21]). Previous studies also reported hippocampal error signals in humans and rodents. For example, intracranially human EEG recording studies reported hippocampal error signals in multiple tasks[45]. Deadwyler et al.[46] applied population analysis to hippocampal CA1 and CA3 recordings while rats performed a two-lever operant version of a spatial delayed-nonmatching-to-sample task. They found that errors contributed to a significant portion of the variance of the population neural activity in both hippocampal CA-fields. In addition to the previous studies, the present study shows not only the error signals in HPC but also provides a direct comparison to EC, and suggests that EC might play a more prominent role than HPC in error detection during error-driven learning.

Beyond the EC and HPC, other studies suggest the involvement of neuromodulatory systems in error detection. For example, serotonin has long been suggested to play a major role in responding to unwanted outcomes and provoke inhibitory responses[47]. Previous studies have shown that serotonergic neurons in the dorsal raphe nucleus signal either the positive or negative reward value[48,49]. The serotonergic cells in these studies resemble the EC-HPC outcome-selective neurons in that they encode reward value with a stable response amplitude. This contrasts with the well-studied, reward-predicting dopaminergic neurons of the ventral tegmental area that respond maximally to unpredicted reward with a gradually diminishing response as the reward contingency becomes more predictable or learned[50,51]. Other studies have shown that acute depletion of serotonin impairs reversal learning, which is mainly based on negative feedback information[52]. In addition, the dorsal raphe nucleus sends extensive projections to both EC and HPC[53], and evidence shows that the serotonergic inputs to the MTL influence learning performances. All 18 types of serotonin receptors are expressed in HPC, and pharmacological manipulation of different types of receptors highly influences the learning behavior in different manner[54,55]. These findings suggest that the error-related or reward-value-related information originating from the dorsal raphe nucleus may influence the error signals observed in the EC and HPC.

**Assessing the effect of different learning strategies on neural activity.** One key realization we made during the course of our analysis is that the monkeys used in our studies were not using the same learning strategy across the two different associative learning tasks we have used in the lab[21,32,56,57]. First, we found clear evidence that the LST animals used an error-driven learning strategy defined as better behavioral performance after error trials relative to after correct trials. This led us to ask which learning strategy animals were used during an object-place associative learning task (OPT) in which we also had extensive hippocampal recordings. To our surprise, in this latter task, animals used a correct-driven learning strategy (i.e., better performance after correct relative to after error trials). We report that for the error-driven learning task (LST), we saw a clear population selectivity shift during the early stage of learning in HPC (Fig. 2a and Supplementary Fig. 5d), that was not seen during OPT where animals used a correct-driven learning strategy (Figs. 3a, d and 6f). Differential neural signals associated with distinct behavioral strategies were also reported in a study in inferotemporal cortex (IT) showing striking shifts in recognition-related signals with a subtle shift in task demand which required a different behavioral strategy[58]. Taken together our results suggest that the hippocampus plays differential roles in error-driven relative to correct-based learning. Future studies comparing the

neural signals in the EC across both types of learning strategies will be of interest.

**Timing of new associative learning in HPC and EC.** One of the questions we were most interested in was comparing and contrasting the time course of error-driven learning-related signals between the HPC relative to the EC (Fig. 6). Consistent with predictions by Lorinz and Buzsaki[24] that the earliest selectivity shift would take place in HPC, we found enhanced stimulus-selectivity following error relative to correct trials in the hippocampus before behavioral learning criterion was reached (Figs. 2a, 6d and Supplementary Fig. 3a), but no such early learning signals in the EC (Figs. 2c, 6f and Supplementary Fig. 3c). A finding by Li et al.[59] also supported early learning signals in the rodent hippocampus, though parallel recordings in the EC were not done. In that study, Li et al. monitored activity in the mouse hippocampus as they performed an odor-based associative learning task. Using a combination of optogenetic and electrophysiology, they showed that hippocampal pyramidal neurons acquired olfactory selectivity before the animals reached learning criteria and that selectivity continued to increase as animals continued to learn. Two additional learning-related signals were seen in the present study, one in the HPC and the second in the EC. In HPC, nearly 20% of the recorded neurons increased or decreased their firing rates in parallel with learning (the changing cells, Figs. 1b, 5a, b and 6e) as we have reported before[32]. By contrast, in EC, the error-up cells increased their stimulus-selectivity after learning relative to before learning (Fig. 4a and Supplementary Figs. 5, 6f). Igarashi et al.[60] also reported cells in both dorsal CA1 and lateral EC acquired odor-selectivity as rats learned to associate an odor-cue to a specific location over 3 days of training, though in that study they reported the lateral EC selectivity shifts occurred slightly earlier than HPC. Task and species differences between the present study and the Igarashi study may underlie the difference in timing of the associated learning signals reported by the two studies.

Figure 6 summarizes the time course of both the error-detection signals and the various learning-related signals we observed during the LST. These findings suggest a strong interplay between the EC and HPC during new learning and future studies doing simultaneous recordings in both MTL areas during associative learning will be essential to further specify the nature of these interactions.

**Conclusion**
Many computational models have hypothesized that the MTL is critical for error-driven learning[61–63], however, few behavioral physiology studies have been done to directly characterize predictions from these models. Here we identify an associative learning task in which animals used a clear error-driven learning strategy to characterize, compare and contrast the neural signals in the EC and HPC. While it is not surprising that EC and HPC are involved in associative learning, this study highlights the prominent error-detection signals and the strong error-related learning signals in the EC. Similarities of the EC error-detection signals to those described in the ACC[13,35] suggests a more prominent functional link between the EC and the classical error-detection network centering at the ACC than previously appreciated. But this is not the first time that the EC and ACC have been functionally linked. Many previous studies have studied the relationship between these two structures in long-term consolidation[64–66]. The findings we report here suggest that connections between the EC and ACC[39–42] are not only involved in the consolidation of long-term memories but those connections may also be involved in the earliest stages of

new long-term associative learning through their strong error-detection signals. Following the time course of the interactions between these two areas from the very first trial of learning through long-term consolidation will be of great interest in future studies.

## Methods

**Subjects**. All procedures and treatments were done in accordance with NIH guidelines and were approved by the NYU animal welfare committee. Six male and one female macaque monkeys were used in this study. The data sets acquired from the entorhinal cortex (EC) and hippocampus (HPC) of monkey A (rhesus, 11.5 kg) and B (bonnet, 7.8 kg) were previously described in Hargreaves et al.[22]. The data acquired in the HPC of monkey A (the same monkey as EC recording but in different sessions) and C (rhesus, 13 kg) were from the same data set as described in Wirth et al.[32]. The hippocampal neurons recorded in monkey EI (rhesus, 6.0 kg) have not been previously published. The data set from Wirth et al.[32] was only used for calculating the stimulus-selectivity shift but not for calculating the outcome-selective signals because the neural activity during inter-trial intervals (ITI) were not recorded. The data acquired in the HPC of monkey M (rhesus, 14.2 kg) and Er (bonnet, 6.3 kg) performing the object-place task was the same data set as in Wirth et al.[21].

**Recording and surgery**. The recording and surgical methods have been described previously[21,22,32]. The animals were implanted with a headpost prior to their behavior training. After they were familiar of the tasks, a recording chamber was placed stereotaxically during the surgery after identifying the recording sites in each of the animals' brain using magnetic resonance imaging (acquired prior to the implantation surgery). The same images were used to identify the recording sites. The electrodes used include single tungsten electrodes (Epoxylite insulation, FHC, USA), glass-coated tungsten electrodes (Alpha-Omega, Israel), or tetrodes (platinum-tungsten, quartz insulated, Thomas Recording, Germany). The electrodes were inserted into the brain through a stainless guide-tube (23 G) positioned in a grid-recording system (Crist Instrument, USA).

The recording sites of newly recorded hippocampal neurons in monkey A, B, and El were plotted in Supplementary Fig. 1a. The recording sites in HPC of monkey A and C performing LST (where activity during the inter-trial intervals was not recorded) are shown in Wirth et al.[32]. The recording sites in HPC of monkey M and E performing OPT were shown in Wirth et al.[21]. The recording sites of EC neurons were plotted in Supplementary Fig. 1b. The recording sites in the hippocampus in monkey A ranged from AP 8 to 19, in monkey B from AP 8 to 12, in monkey C from AP 8 to 19, in monkey El from AP 8 to18, in monkey M from AP 11 to 21 and in monkey E from AP 6 to 16. The recording sites in EC in monkey A ranged from AP 12 to 16, ML 9 to 12, and in monkey B from AP 13 to 16, ML 7.5 to 12. The coordinates are all calculated from Frankfurt zero (the crossing of interaural line and midline of the brain) in mm and listed in Supplementary Data 1.

To address the possibility that the effects described in the present study could be explained by the specific anatomical location of the neurons, we performed correlation analysis between the selectivity indices and the recording sites along three dimensions: AP, ML, and relative depth dimensions. In EC, we found no significant correlation between the differential selectivity of error-up, correct-up, or non-outcome-selective neurons. Similarly, in hippocampus, we found no significant correlation between the recording sites and the selectivity either.

## Behavioral tasks

*Location-scene task (LST, Fig. 1a)*. The LST has been used extensively in previous studies from our laboratory[22,32,56,57]. Briefly, each trial starts with a baseline period where animals were required to fixate a central fixation spot (300–750 ms; Fig. 1a). Then four targets superimposed on a novel, natural, colorful scene were presented (500–750 ms). After a delay (700–1000 ms) the fixation spot disappeared, cueing the animals to make a saccade to one of the four targets. Fixation was required from the beginning of the baseline period until they were cued to make an eye movement, and only one of the targets was rewarded for each visual scene. Each day 2–4 novel scenes were presented and animals learned to associate each new scene with a specific rewarded target location through trial and error. Randomly intermixed in with the new scenes were 2–4 highly familiar "reference" scenes presented to control for the eye movement or reward-associated neural activity.

*Object-place task (OPT, Fig. 3a, b)*. This task has been described in detail before[21]. Here, monkeys learned to associate an object-place combination to one of two possible bar-release responses (early vs. late) to obtain rewards. The animals initiated each trial by holding a bar and fixating a central fixation spot (Fig. 3a). After a 500 ms baseline period one of the two novel objects was shown for 500 ms at one of the two locations (that changed daily) on the screen (4 combinations total). After a 700 ms delay the animals could release a bar either during the 500 ms presentation of an orange cue stimulus ("early" release) or continue holding until a green cue stimulus was presented (for 500 ms) immediately afterward to make a "late" release. If the response was correct, an auditory feedback tone was played, and after a random delay (30–518 ms) 2–4 drops of juice were delivered as rewards.

The animals were required to fixate from the beginning of the baseline period until the early or late bar release. The associations between the object-place combinations to the bar-release responses were counterbalanced as illustrated in Fig. 3b.

**Data analysis**. All data analysis was done with custom-written Matlab programs (Mathworks, Natick, MA, USA). The outcome-related criteria and selectivity index analysis were the same as a previous publication[21] with several modifications to accommodate the present data set. The same as the previous study, the baseline period was defined as the 300 ms before the scene onset and the scene period was the 500 ms duration from the onset of the scene. Slightly different from the previous study, the delay period was defined from scene offset to 600 ms afterward (instead of the entire 700 ms period). The outcome period started at the end of the subject's 30 ms fixation of the chosen target (after making a saccade from the central fixation spot) and continued for the following 2000 ms. We used only 600 ms instead of the full 700 ms delay period in the calculation of the scene selectivity index because another study[67] using a similar behavioral paradigm reported eye-movement direction-related changes of cortical spiking activity during the 100 ms right before the eye movement so we did not include this time period in our analysis.

**Estimating learning performance**. We defined whether learning took place and the trial at which learning occurred using a dynamic logistic regression algorithm as described in Writh et al.[21,32]. Learning sessions were defined as those in which animals made at least seven consecutive correct responses for at least one location-scene association. To estimate the learning curve, we parameterized the performance of each trial into a binary sequence ("1" for correct responses and "0" for incorrect responses). We then constructed the learning curve with this binary performance sequence together with 95% confidence bounds using a Bayesian state-space model[68]. The behavior learning trial was defined as the first trial that the lower 95% bound of the estimated learning curve crossed the random choice threshold and stayed above the threshold for the next three trials. The random choice threshold was defined according to how many targets were given in each session. For example, for 4-target LST the threshold was 0.25 and for 3-target LST the threshold was 0.33. The analysis codes for estimating the behavior are available.

The animals performed at least 200 trials each day even when they reached learning criteria early in the session. For those sessions when the animals learned at least one location-scene association, 75% of error trials happened before the animals finished 57% of the trials on average. During the second half of the sessions, the animals made significantly fewer mistakes than the first half ($p < 10^{-6}$).

**Estimating learning strategy**. We determined if animals used either an error-driven or a correct-driven learning strategy by asking whether they performed better immediately after error trials (post-error trials) compared to after correct trials (post-correct trials). In a learning session containing in total $n$ trials ($n$ was typically ~200), there were $p$ error trials and $q$ correct trials ($n = p + q$). If the vector $P_{(1,p)}$ denoted all the error trials in this learning session and the vector $Q_{(1,q)}$ denoted all the correct trials, p was usually much smaller than $q$ (99% of the cases). After parameterized the performance of each trial into binary series ("0" indicated error and "1" indicated correct trials), we compared the performance of $(P_{(1,p)} +1)$th trials (post-error trials) and $(Q_{(1,q)} +1)$th trials (post-correct trials) using a two-tail $t$-test. We discarded the last error trial if $P_{(p)} = n$. If the performance of post-error trials was significantly higher than post-correct trials, we defined this as an error-driven strategy to learn. If the performance of post-correct trials was significantly higher than post-error trials, this was defined as a correct-driven strategy.

For these calculations, we used only the performance of the first $p$ (i.e., the total number of error trials) post-correct trials and discarded the $(Q(p + 1,q) + 1)$ th post-correct trials to compare with that of the same number of post-error trials ($(P_{(1,p)} + 1)$th trials). As we mentioned previously, this is because the animals performed at least 200 trials for each session, even though they have usually reached learning criteria (as defined in LST session) after about 100 trial. For about the second half of each session, the animals were usually performing at ceiling with very few if any error trials. Thus, our calculations for defining either an error-driven- or a correct-driven- learning strategy were based exclusively on this early stage of learning using the same number of correct and error trials.

**Outcome-selective signals**. The outcome-selective signals (correct-up and error-up cells) were defined based on a previous publication[21] with two slight modifications (see below). As in the previous study, we analyzed spiking activity from the end of the 30 ms fixation of the target (after making a saccade from the central fixation spot) and for the following 2000 ms of the inter-trial interval (ITI). We calculated the mean firing rate during the first and second 1000 ms of the ITI interval separately, normalized by subtracting the mean firing rate during the baseline period, and compared whether the neurons responded significantly more to correct or error outcome during either of these periods by performing paired $t$-test. To correct for multiple comparisons the significance threshold was set to $p < 0.025$. In our previous study[21], we separately reported neurons that increased their firing rates during either half of the ITIs compared to baseline after correct trials as

correct-up cells and those that decreased their firing rates after error trials as error down cells. Because there was no error down cells in either EC or HPC while animals performed LST, the first difference with previous studies[21] was that correct-up cells were defined as those cells whose normalized mean firing rate in either of the two halves of the ITIs after correct trials were significantly higher than after error trials (first difference). Also, slightly different from previous studies[21], here error-up cells in this study were defined as the normalized mean firing rate during either half of the ITI after error trial was higher than correct trials, regardless of where the difference came from. All error-up neurons in HPC increased their firing rates during either half of the ITIs compared to baseline after error trials. Of 43 error-up cells in EC, 37 neurons increased their firing rates after error responses compared to baseline, and six neurons decreased their firing rates after correct responses.

In a previous study, we found that neither correct-up nor error-up cells changed the magnitude of their neural responses over the course of learning OPT[21]. Here we also examined whether the amplitude of the neural responses of the outcome-selective cells changed over the course of learning the LST. As in the previous study, we used one-way ANOVA with the time period (early, middle and late period of the session) as the main factor to examine the amplitude of the correct-up and error-up signals over time. We analyzed the first 1000 ms and the second 1000 ms after the saccade separately and found no difference in the amplitude of the correct-up and error-up signals.

**Defining changing cells.** In a previous publication using the same LST, we defined a population of changing cells in the HPC that signaled new associative learning by changing their stimulus-selective response correlated with learning[32]. We followed the same criterion described in Wirth et al.[32] to identify changing cells in the HPC and EC in this study. Briefly, for those neurons recorded during successful learning sessions, we extracted the raw firing rates of the cells during the scene and delay period then correlated the trial-by-trial mean firing rates with the estimated learning curves in the same session. Those cells significantly correlated with the learning curves were defined as changing cells, and the significance level was set to $p < 0.025$ to correct for multiple comparisons. Consistent with the previous study, we found two categories of changing cells both in the HPC (Fig. 5a, b) and EC (Fig. 5c, d). Sustained changing cells increased their firing rate correlated with the animal's behavioral learning curve while baseline sustained changing cells typically decreased their firing rate anti-correlated with behavioral learning.

**Selectivity index.** The selectivity index (SI) was calculated as in Wirth et al.[21]. We extracted the normalized mean firing rates of scene and delay periods by subtracting the mean baseline firing rate for each individual neuron. The following equation was used to calculate the selectivity index:

$$SI = \left( n - \sum_{i=1}^{n} \left( \frac{\lambda_i}{\lambda_{max}} \right) \right) / (n - 1),$$

where $n$ was the total number of location-scene combinations, $\lambda_i$ was the normalized mean firing rate of the neuron to the $i$th combination and $\lambda_{max}$ was the normalized maximum firing rate of all the combinations. The SI was calculated for each individual neuron before and after the monkeys learned the first scene-location association to the criteria. To estimate the sensitivity of the SI, we have performed the permutation test by permuting the labels of the scenes 1000 times and took the 95% quantile as the threshold to select those neurons whose SI surpasses the threshold. The performance of learning was estimated by the Bayesian State-Space Model as used in our previous studies[21,32,56]. After the differential SI (defined as SI after – SI before learning) was obtained for each individual neuron, a one-sample $t$-test was applied to population differential SI for each group of neurons to estimate whether the neurons changed their SI depending on learning.

**Area under receiver–operating characteristic (ROC) curve.** To quantify how well the outcome-selective neurons can differentiate correct from error outcomes, we calculated the time courses of area under receiver–operating characteristic curve (area under ROC) of the mean firing rates during the inter-trial intervals. As the outcome-selective signals, we analyzed spiking activity from the end of the 30 ms fixation of the target (after making a saccade from the central fixation spot) and for the following 2000 ms of the inter-trial interval (ITI). For each outcome-selective neuron, we calculated the ROC curve using the mean firing rates of all correct and error trials every 300 ms, with a sliding step of 50 ms. The area under the ROC curve is then calculated for each neuron over the 2000 ms inter-trial-interval time. The averaged time courses of the area under the ROC curve are then plotted separately for all the correct-up and error-up cells in EC or HPC to show the strength of differentiating correct from error outcomes.

**Statistics and reproducibility.** All data presented in the text are shown adhering to APA style by reporting the $t$-values with a degree of freedom and spelled out all the exact $p$-values beside the cases while $p < 0.001$. Sample sizes are indicated in detail in each figure caption, main text, and corresponding method sessions. Exclusion criteria, if applied, are specified in each corresponding method section.

## Data availability
Data available upon request to S.K.

## Code availability
Accession of codes (Matlab) is available via the Github repository https://github.com/kushihpi/Ku_error_driven_Learning_CommBiol2021.

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

## Acknowledgements

We would like to thank Prof. György Buzsáki for his insightful comments on a previous version of the manuscript; Dr. Hweeling Lee for her helpful discussions in statistics; Ellen Wang and Joyce Ho for their assistance in animal experiments; Shohan Hasan for assistance in task programming and anonymous reviewers for suggestions to improve the manuscript. The work is supported by NIH Grants: R01 MH084964-01 "The functional organization of the medial temporal lobe"; R01- NIMH "Neurophysiological and fMRI studies of associative learning in the MTL and striatum".

## Author contributions

Conceptualization, S.K. and W.A.S.; experiments, S.K., E.L.H., and S.W.; analysis, S.K.; experimental design, S.K., E.L.H., S.W., and W.A.S.; writing/reviewing/editing, S.K. and W.A.S.

## Funding

## Competing interests

The authors declare no competing interests.
