## [Peer Review File · Communications Biology]

Reviewers' comments:

Reviewer #1 (Remarks to the Author):

In this well-written manuscript, Ku and colleagues aim to establish the relative role of hippocampus (HPC) and entorhinal cortex (EC) in providing error-detection signals in error driven learning. They use a collection of electrophysiology data from a set of macaque monkeys. The authors claim that error-detection signals are more prominent in EC relative to HPC and that error-driven learning induces plastic changes in the HPC but not the EC, overall suggesting differential contributions of these two brain areas. While the manuscript is well-rooted in previously published work from the co-authors and the overall experimental approaches are sound, I have trouble accepting the key result from the data presented.

Major point:

- The statistical tests employed make certain assumptions about the data that I do not seem to be able to verify from what is presented in the manuscript. It is unclear precisely where in HPC and EC recordings were obtained. Recordings come from sites many mm apart and thus could be coming from different areas/layers in HPC or EC. It would help to (a) list the approximate location of recording sites per animal (instead of pooled, SuppFig.1) and (b) demonstrate that data from different animals can be pooled without violating the assumptions of independence etc of the statistical tests used. Given what we know, and the literature outside the nHP field should be consulted (and consequently, maybe claims about a lack of in vivo evidence in the abstract moderated), about different spiking activity profiles of neurons in HPC and EC at rest and during learning tasks.

Minor points:

- Please add more quantitative data to the main text results and include appropriate statistical reporting, including actual F and p values.
- The overall sample sizes are very small; would the authors be able to provide estimates of the expected effect sizes? Related to the major point above, could the effect observed in Fig.2 and Fig.4 be simply caused by differences in the populations of neurons sampled?
- On a related note, would the authors be able to add a discussion (or model) on the sensitivity of the selectivity index to differences in the baseline spiking of cells sampled, especially given the overall low spiking rate? How confident can you be in the SI measuring the effect you are reporting?

Reviewer #2 (Remarks to the Author):

The study by Ku et al. uses in vivo electrophysiology in monkeys to show that neuronal activity in EC and HPC reflect the distinct roles of these areas in error-driven learning of novel visual associations. Error detection signals were present in both areas but were stronger and more prevalent in EC compared to HPC. However, error-driven learning was mainly reflected in hippocampal neurons, which exhibited increases in population stimulus-selectivity following error relative to correct trials. The authors also showed distinct activity patterns in EC and HPC after learning, reflecting their distinct roles in long-term memory storage of learned associations. The paper is very well-written and the methods are rigorous, and have been validated and replicated in previous work. The study has important novel contributions to our understanding of how error processing can influence learning and long-term memory associations in primates. A few recommendations for improvement of clarity are listed below:

- 1) Abstract, lines 34-36: "After learning, different populations of cells in both the EC and HPC signaled long-term memory with enhanced stimulus-selective responses." This sentence is a bit vague: "long-term memory " of what (of novel visual associations?)

- 2) Results lines 143-153, Figure 1C-G and Supp Fig 4: It is unclear to me why the HPC cell example in Fig 1G was classified as a correct up cell, given that the plot shows that the firing rate during correct trials did not increase relative to error trials. The same goes for the HPC correct up cell examples in Supp Fig 4. The text also noted, "The AU-ROC of correct up cells in HPC was not

significantly greater than zero (sup. Fig4D, $p > 0.05$, ttest)", hence adding to the confusion. Further, why was Fig 1G not cited with this statement as an example? In general, I am unclear if Figure 1C-G and Supplemental Fig 4 show the same thing – examples of averaged and normalized population firing rates of correct up and error up cells. If so, is it necessary to have both? Please clarify the classification of correct up cells, and the examples given in Figures 1C-G and Supp Fig 4.

3) Results, lines 217-228: I am a bit unclear how changing cells differ/overlap from error-up/correct-up cells. For instance, in Supp Fig 3 the EC example of a "changing cell" also seems like an "error-up" cell to me (from what I understood in figure 1)? It would help if the authors elaborate a bit more here what is meant by "changing cells" with regards to how they change firing rates in relation to the learning curves.

Supplementary Figure 3 should be part of the main figures, as it is one of the main conclusions of the paper. Perhaps combine with Figure 4 to illustrate the contrast between the long-term memory signaling in EC (enhanced selectivity in the error-up cells) vs HP (changing cells).

4) In Supp Fig 3, it looks like the examples of cells were from very different learning curves: The HPC cell was from a set of trials with mostly correct trials, and then the EC was from a set with more error trials. Presumably, wouldn't these two sets will have different degrees of error-driven learning? How do the changing cells behave then in the two areas when error/error-driven ?

5) In Supp Fig 3, both the estimated firing rate and learning curve were plotted on the same plot – but with the y-axis labeled only for the behavioral measure (probability correct). I am unclear about the unit for firing rates on the plot – I assume this is represented as normalized to baseline? If so the legend should say this, and a second set of axis labels for the firing rate units should be made (to the right of the plot), for clarity.

6) Fig 3, It would help if the authors present a trace similar to Fig. 1B to clearly contrast between the two learning strategies.

7) Fig 4. It would be helpful to also see some of the scatter plots, similar to Figure 2.

8) Discussion, lines 267-272, 355: As the authors noted "Further studies involving simultaneous recording the ACC and EC will be required to determine the directionality and functional interconnectivity between these regions" (lines 267-272). However, expanding the discussion on ACC- EC, ACC-HPC and EC-HPC connectivity/flow of activity and to speculate a bit how error processing can be used in memory will be interesting and helpful. For instance, Paz/Pare work on how mPFC inputs can facilitate information flow from EC to HPC (Paz et al., 2007) , or the Barbas work showing synaptic pathways from ACC terminating in EC directly (e.g., Bunce et al, 2011; Joyce & Barbas, 2018); as well as to note, in contrast to the EC-ACC bidirectional pathways, the largely unidirectional HPC outputs to ACC (Barbas & Blatt, 1995). Moreover it is important to note and discuss that error activity in ACC occurs earlier (within < 500 msec; e.g., Niki & Watanabe, 1979; Amiez et al, 2005) after the motor response error, compared to the EC cells shown here, especially with the statement describing "Similarities of the EC error-detection signals to those described in the ACC" (line 355).

Paz, R., Bauer, E. P., & Pare, D. (2007). Learning-related facilitation of rhinal interactions by medial prefrontal inputs. *J Neurosci*, 27(24), 6542-6551. doi:10.1523/JNEUROSCI.1077-07.2007

Joyce, M. K. P., & Barbas, H. (2018). Cortical Connections Position Primate Area 25 as a Keystone for Interoception, Emotion, and Memory. *J Neurosci*, 38(7), 1677-1698. doi:10.1523/JNEUROSCI.2363-17.2017

Bunce, J. G., & Barbas, H. (2011). Prefrontal pathways target excitatory and inhibitory systems in memory-related medial temporal cortices. *Neuroimage*, 55(4), 1461-1474

Barbas, H., & Blatt, G. J. (1995). Topographically specific hippocampal projections target functionally distinct prefrontal areas in the rhesus monkey. *Hippocampus*, 5, 511-533.

Niki H, & M Watanabe (1979) Prefrontal and cingulate unit activity during timing behavior in the monkey. *Brain Res.* 1979 Aug 3;171(2):213-24. doi: 10.1016/0006-8993(79)90328-7.

Amiez C, Joseph J-P, and Procyk E (2005), Anterior cingulate error-related activity is modulated by predicted reward *Eur J Neurosci.* 2005 Jun; 21(12): 3447–3452.

9) Discussion. line 231: The statement, "Here we showed that EC and HPC play prominent but distinct roles in error-driven learning" is little too strong and slightly inaccurate. As with in vivo electrophysiological studies, the data in the current study are correlative and do not show causation per se. Perhaps it is more accurate to state that the data importantly showed that "neuronal activity in EC and HPC reflected the important but distinct roles of these areas in error-driven learning."

Replying to the reviewers comments:

The contributions of entorhinal cortex and hippocampus to error driven learning

Blue text: reviewer's comments

Black text: author's reply

Reviewer 1:

In this well-written manuscript, Ku and colleagues aim to establish the relative role of hippocampus (HPC) and entorhinal cortex (EC) in providing error-detection signals in error driven learning. They use a collection of electrophysiology data from a set of macaque monkeys. The authors claim that error-detection signals are more prominent in EC relative to HPC and that error-driven learning induces plastic changes in the HPC but not the EC, overall suggesting differential contributions of these two brain areas. While the manuscript is well-rooted in previously published work from the co-authors and the overall experimental approaches are sound, I have trouble accepting the key result from the data presented.

Major point:

- The statistical tests employed make certain assumptions about the data that I do not seem to be able to verify from what is presented in the manuscript. It is unclear precisely where in HPC and EC recordings were obtained. Recordings come from sites many mm apart and thus could be coming from different areas/layers in HPC or EC. It would help to (a) list the approximate location of recording sites per animal (instead of pooled, SuppFig.1)

While we agree with the reviewer that exact laminar and sub-region specificity for all our recording sites would be valuable, unfortunately similar to the vast majority of the behavioral neurophysiological approaches in monkeys, that level of detail is simply beyond the sensitivity of our current recording techniques. As much as the reviewer has pointed out we certainly find it extremely interesting and important to know more detailed information about the recording sites in any study. However, exact layer information of electrophysiological recording can only be obtained by other recording techniques such as utilizing laminar probes combined with current source density analysis, which is beyond the scale of the present study. With the single electrodes or tetrodes we can only know the

relative recording depth and still do not know in which exact cortical layers the neurons were recorded. Localization of recording sites in the experiments in the present study can only be approximate and laminar specificity in the EC, for example is beyond the resolution of the approach. As shown in sup. Fig. 1B, the relative depths were actually represented. Within every grid, each circle indicated the recording site of one neuron. The closer to the upper boundary of the grid, the higher the recording site (relative to the surface of the brain from the recording chamber) was, i.e. the more possible the neurons were localized in layer 5/6 in EC (Note that EC is located on the bottom of the monkey brain, meaning that the superficial layer faces toward the bottom, i.e. the depth is lower relative to the EC deep-layers.). In contrast, the closer to the lower boundary of the grid the more possible they were to be in layer 2/3. Also, while we are very familiar with the detailed cytoarchitectonic subdivisions of the monkey entorhinal cortices¹, precisely identifying the sub-area within the EC is also beyond the scope of our techniques. We can say that we have recorded from the 12-16mm AP and 8-12mm ML that likely included the rostral, intermediate and caudal subdivisions of the EC.

For the hippocampus, while the electrophysiological signal made it clear that we had entered the hippocampus, the precise layer/subdivision of the hippocampus where the final recording was made is also difficult to confirm. Given the small size of the DG granular cells and general difficulty of isolating them, we estimate that the vast majority of our recordings were made in the pyramidal layers of CA3 and CA1 in the middle 2/3 of the body of the hippocampus.

The goal of the present study is to show to what extent EC and hippocampus contribute to error driven learning as first *in-vivo* evidence in the literature. If our study can provoke further interests and effort of more detailed investigation such as further studying about the contributions and functional interactions of different subregions or cortical layers to error driven learning in medial temporal lobe, that would be extremely rewarding outcomes for the authors. Nevertheless we did 1) listed the range of recording in each subject in line 445-450 and 2) carefully documented the relative recording depths of EC as well as the newly recorded hippocampal neurons, i.e. at the same AP-ML location we did have the information of which neurons were located at deeper or superficial layers. Based on this information we have applied the correlation analysis to show 1) whether there is a linear correlation between the relative depth and the outcome selectivity and 2) whether there are layer-

differences between stimulus selectivity shifts before versus after learning and after correct versus error trials. We will discuss the findings in the later paragraphs related to the reviewer's third minor point. In short, we did not find any significant relationship between the recording sites within EC or hippocampus and the selectivity discussed in the present study.

(b) demonstrate that data from different animals can be pooled without violating the assumptions of independence etc of the statistical tests used. Given what we know, and the literature outside the nHP field should be consulted (and consequently, maybe claims about a lack of in vivo evidence in the abstract moderated), about different spiking activity profiles of neurons in HPC and EC at rest and during learning tasks.

To address this point, we ensured that the within-subject variances were not different from between-subject variances in order not to violate the assumptions of independence of different categories (i.e. different subjects). To demonstrate that the data in the present study fulfills this criterion, we performed the Pearson *Chi*-square independence test on the firing rates across all included subjects and the $\chi^2(165) = 1394.5, p < .001$ in hippocampus. In EC, $\chi^2(178) = 242.4, p < .001$. In addition, we separately plotted the selectivity index (SI) of each individual animal in sup. Fig. 4 and 5 to further demonstrate that not only the firing rates, the variances of calculated SI within subjects were not different between subjects either. The test results were included in the legend of sup. Fig. 4 and 5.

Similar experimental/statistical approaches design (i.e., repeated measure in the same subject) are common in behavioral electrophysiology studies using either NHP or rodents in hippocampus²⁻⁷, entorhinal cortex^{8,9}, prefrontal cortices¹⁰⁻¹² and visual cortices¹³⁻¹⁶.

Minor points:

1) Please add more quantitative data to the main text results and include appropriate statistical reporting, including actual F and p values.

We have expanded the statistical report throughout the whole paper adhering to APA style by reporting the *t*-values with degree of freedom and spelled out all the exact *p* values beside the cases while $p < .001$. We also added the Z-values in addition to *p*-values while

using Wilcoxon signed-rank test. We usually use *t*-test throughout the manuscript because we have designed the experiments to have a paired control-condition. By contrast, we did not use ANOVA in the present study, so we did not have any *F*-values to report.

2) The overall sample sizes are very small; would the authors be able to provide estimates of the expected effect sizes?

We have added the Cohen's *d* for all the two-sample *t*-tests and demonstrated that all the Cohen's *d* ranged from 0.2 up to 10, indicating that the effect sizes were at least medium and usually large. We also performed the power analysis for both EC and hippocampus (see figures below). Based on the conventional statistical power level¹⁷ (80%), for detecting the differences between the post-correct versus post-error selectivity indices in hippocampus the minimum sample size is 35 (labeled with a green circle). For EC, the minimum sample size to detect the difference between the selectivity before versus after learning is 31 to achieve 80% statistical power. Our sample sizes of error-up cells in EC (42) and all recorded hippocampal neurons (168, 102 after corrected by permutation test) both exceeded the minimum sample sizes calculated accordingly (labeled with red circles), suggesting our experiment design to be valid.

3) Related to the major point above, could the effect observed in Fig.2 and Fig.4 be simply caused by differences in the populations of neurons sampled?

We tackle this issue in two parts: first, we discuss the possibility whether the effect could be explained by any anatomical tendency of the neurons. Second, we discuss whether the effect could be due to the data recorded in only one single subject.

For the first concern, we performed correlation analysis between the selectivity indices and the recording sites along three dimensions: AP, ML and relative depth dimensions. In EC, we found no significant correlation between the differential selectivity of error-up, correct-up or non-outcome selective neurons. Similarly, in hippocampus we found no significant correlation between the recording sites and the selectivity either.

For the second concern, we plotted the selectivity indices of individual animals separately in supp. Fig. 4 and 5, visually demonstrating that the distributions of selectivity indices (SI) of different animals were similar across feature spaces. We also applied the *Chi*-square test of homogeneity and can show that we cannot reject the null-hypothesis that the variances of each set of individual animal is from the same population and therefore we can pool the data of individual animals together. This is described in the paragraph in lines 188-194.

4) On a related note, would the authors be able to add a discussion (or model) on the sensitivity of the selectivity index to differences in the baseline spiking of cells sampled, especially given the overall low spiking rate? How confident can you be in the SI measuring the effect you are reporting?

To address this possibility we added the following procedure to ensure the robustness of SI-calculation : the firing rates were first normalized to baseline firing of each neuron to avoid the potential confront of different baseline firing rates. We always set a minimum-trial criterion so that the SI could be obtained robustly not by a single trial activity but at least 3 repetitions. In addition, we have also confirmed the sensitivity of the selectivity index (SI) by performing permutation test to the preliminary data but did not report it in the last version of the manuscript. Now we included that permutation test procedure in the Method session (line 601-604) to demonstrate the sensitivity of the SI-calculation. Briefly, we permuted the labels 1000 times and calculated the SI based on the permuted labels. Then we used only those neurons whose correct SI lays beyond the threshold (5%) of the permutation test to show the differences after correct or error trials in hippocampus and the changes after learning in entorhinal cortex. The result is, the SI of only three entorhinal error-up cells did

not surpass the threshold, and the SI of 102 out of 168 hippocampal neurons did surpass the threshold. Since the SI of some hippocampal neurons did not surpass the threshold, we took those SI after correct versus after error trials of the neurons with higher-than-threshold SI and found out that the differences in SI between post-correct and post error trials were not only significant but also seem to be even larger. As shown in sup. Fig. 2A, using the paired *t*-test, the $t(101) = 2.79$, $p = .0062$, $d = 0.2215$. All new results are described in lines 182-187.

Reviewer 2:

1) Abstract, lines 34-36: “After learning, different populations of cells in both the EC and HPC signaled long-term memory with enhanced stimulus-selective responses.” This sentence is a bit vague: “long-term memory “ of what (of novel visual associations?)

The sentence has been changed from ‘After learning, different populations of cells in both the EC and HPC signaled long-term memory with enhanced stimulus-selective responses.’ to ‘After learning, different populations of cells in both the EC and HPC signaled long-term memory of newly learned associations with enhanced stimulus-selective responses.’

2) Results lines 143-153, Figure 1C-G and Supp Fig 4: It is unclear to me why the HPC cell example in Fig 1G was classified as a correct up cell, given that the plot shows that the firing rate during correct trials did not increase relative to error trials. The same goes for the HPC correct up cell examples in Supp Fig 4. The text also noted, “The AU-ROC of correct up cells in HPC was not significantly greater than zero (sup. Fig4D, $p > 0.05$, *t*test)”, hence adding to the confusion. Further, why was Fig 1G not cited with this statement as an example? In general, I am unclear if Figure 1C-G and Supplemental Fig 4 show the same thing – examples of averaged and normalized population firing rates of correct up and error up cells. If so, is it necessary to have both? Please clarify the classification of correct up cells, and the examples given in Figures 1C-G and Supp Fig 4.

The time courses plotted in the original version fig. 1 C-G (fig. 1 E,G,I,K in the revised version) and original supp. Fig. 4 (now sup. Fig. 2) are averaged time courses and AU-under ROC of all error-up or correct up cells in either EC or HPC. Even though the difference between correct and error trials of each individual neuron is significant, the averaged time courses across all

outcome selective neurons may show overlapping time courses as a population. Therefore, fig. 1G in the original version (fig. 1K for the revised version) seems not to have clear difference between correct and error trials. To avoid confusion, the time courses of one example neuron for each condition are added to fig. 1. Specifically, fig. 1D shows the time courses of one representative error-up cell in EC, fig. 1F shows the time courses of one representative correct-up cell in EC, fig. 1H shows the time courses of one representative correct-up cell in HPC, fig. 1J shows the time courses of one representative correct-up cell in HPC.

3) Results, lines 217-228: I am a bit unclear how changing cells differ/overlap from error-up/correct-up cells. For instance, in Supp Fig 3 the EC example of a “changing cell” also seems like an “error-up” cell to me (from what I understood in figure 1)? It would help if the authors elaborate a bit more here what is meant by “changing cells” with regards to how they change firing rates in relation to the learning curves.

The changing cells are defined as “the trial-by-trial averaged firing rates of the cells during the scene or delay period are significantly correlated with the estimated learning curve” (method session 4D, ³). We apologize for the confusion due to mixing color schemes. Now we changed the presentation of estimated learning curve in fig. 1B, fig. 3C and originally supp. Fig. 3, now fig. 5 to teal color and the estimated trial-by-trial firing rates of the changing cells in previous supp. Fig. 3 now fig. 5 to purple color so that they are not confused with the correct or error outcome presentation in blue and red.

Supplementary Figure 3 should be part of the main figures, as it is one of the main conclusions of the paper. Perhaps combine with Figure 4 to illustrate the contrast between the long-term memory signaling in EC (enhanced selectivity in the error-up cells) vs HP (changing cells).

The discovery of hippocampal changing cells was reported in detail in Wirth et. al. ³ and therefore we did not include the changing cell presentation in the main body of the previous version of the manuscript. However, we agree with the reviewer that including the plots of changing cells does facilitate the understanding of the paper. Now we moved the changing

cell results from the supplementary material to the main text as figure 5.

4) In Supp Fig 3, it looks like the examples of cells were from very different learning curves: The HPC cell was from a set of trials with mostly correct trials, and then the EC was from a set with more error trials. Presumably, wouldn't these two sets will have different degrees of error-driven learning?

To address the reviewer's point, we changed our example cells to one with a similar learning pattern (now fig. 5A).

How do the changing cells behave then in the two areas when error/error-driven ?

Within the 15 EC changing cells, only two are correct-up and three are error-up cells, showing that there was little overlap between the learning signals, i.e. the changing cells rarely encoded outcome selectivity in EC. Similarly, in hippocampus, among the 23 changing cells, only three are correct-up cells and three others are error-up cells. We have observed this kind of rather exclusive property previously and did not report this in the previous manuscript. Now we include these results in the same paragraph describing changing cells (line 256-260).

5) In Supp Fig 3, both the estimated firing rate and learning curve were plotted on the same plot – but with the y-axis labeled only for the behavioral measure (probability correct). I am unclear about the unit for firing rates on the plot – I assume this is represented as normalized to baseline? If so the legend should say this, and a second set of axis labels for the firing rate units should be made (to the right of the plot), for clarity.

We appreciate the suggestion of the reviewer and changed the labeling to match the color scheme of the dual presentation: the estimated learning curves are labeled in teal color, and the corresponding axis at the left side is also in teal. The estimated trial-by-trial firing rates are now in purple, and the corresponding axis at the right side is also in purple.

6) Fig 3, It would help if the authors present a trace similar to Fig. 1B to clearly contrast between the two learning strategies.

One example estimated learning curve was added to fig. 3C.

7) Fig 4. It would be helpful to also see some of the scatter plots, similar to Figure 2.

The scatter plots of correct-up, error-up and non-outcome selective neurons of EC are added to fig. 4 A,B,C. And those of HPC are added to fig. 4F,G,H.

8) Discussion, lines 267-272, 355: As the authors noted “Further studies involving simultaneous recording the ACC and EC will be required to determine the directionality and functional interconnectivity between these regions” (lines 267-272). However, expanding the discussion on ACC- EC, ACC-HPC and EC-HPC connectivity/flow of activity and to speculate a bit how error processing can be used in memory will be interesting and helpful. For instance, Paz/Pare work on how mPFC inputs can facilitate information flow from EC to HPC (Paz et al., 2007) , or the Barbas work showing synaptic pathways from ACC terminating in EC directly (e.g., Bunce et al, 2011; Joyce & Barbas, 2018); as well as to note, in contrast to the EC-ACC bidirectional pathways, the largely unidirectional HPC outputs to ACC (Barbas & Blatt, 1995). Moreover it is important to note and discuss that error activity in ACC occurs earlier (within < 500 msec; e.g., Niki & Watanabe, 1979; Amiez et al, 2005) after the motor response error, compared to the EC cells shown here, especially with the statement describing “Similarities of the EC error-detection signals to those described in the ACC” (line 355).

Paz, R., Bauer, E. P., & Pare, D. (2007). Learning-related facilitation of rhinal interactions by medial prefrontal inputs. *J Neurosci*, 27(24), 6542-6551. doi:10.1523/JNEUROSCI.1077-07.2007

Joyce, M. K. P., & Barbas, H. (2018). Cortical Connections Position Primate Area 25 as a Keystone for Interoception, Emotion, and Memory. *J Neurosci*, 38(7), 1677-1698. doi:10.1523/JNEUROSCI.2363-17.2017

Bunce, J. G., & Barbas, H. (2011). Prefrontal pathways target excitatory and inhibitory systems in memory-related medial temporal cortices. *Neuroimage*, 55(4), 1461-1474

Barbas, H., & Blatt, G. J. (1995). Topographically specific hippocampal projections target functionally distinct prefrontal areas in the rhesus monkey. *Hippocampus*, 5, 511-533.

Niki H, & M Watanabe (1979) Prefrontal and cingulate unit activity during timing behavior in

the monkey. *Brain Res.* 1979 Aug 3;171(2):213-24. doi: 10.1016/0006-8993(79)90328-7.

Amiez C, Joseph J-P, and Procyk E (2005), Anterior cingulate error-related activity is modulated by predicted reward *Eur J Neurosci.* 2005 Jun; 21(12): 3447–3452.

We appreciate the insightful comments of the reviewer and have added text covering these relevant points to of the discussion in line 303-313 about these comments.

9) Discussion. line 231: The statement, “Here we showed that EC and HPC play prominent but distinct roles in error-driven learning” is little too strong and slightly inaccurate. As with in vivo electrophysiological studies, the data in the current study are correlative and do not show causation per se. Perhaps it is more accurate to state that the data importantly showed that "neuronal activity in EC and HPC reflected the important but distinct roles of these areas in error-driven learning."

We have changed the line accordingly (now line 266-267).

Reference

- 1 Suzuki, W. A. & Amaral, D. G. Topographic organization of the reciprocal connections between the monkey entorhinal cortex and the perirhinal and parahippocampal cortices. *The Journal of neuroscience : the official journal of the Society for Neuroscience* **14**, 1856-1877 (1994).
- 2 Wirth, S. *et al.* Trial outcome and associative learning signals in the monkey hippocampus. *Neuron* **61**, 930-940, doi:10.1016/j.neuron.2009.01.012 (2009).
- 3 Wirth, S. *et al.* Single neurons in the monkey hippocampus and learning of new associations. *Science* **300**, 1578-1581, doi:10.1126/science.1084324 (2003).
- 4 Yanike, M., Wirth, S., Smith, A. C., Brown, E. N. & Suzuki, W. A. Comparison of associative learning-related signals in the macaque perirhinal cortex and hippocampus. *Cerebral cortex* **19**, 1064-1078, doi:10.1093/cercor/bhn156 (2009).
- 5 O'Keefe, J. & Dostrovsky, J. The hippocampus as a spatial map. Preliminary evidence from unit activity in the freely-moving rat. *Brain research* **34**, 171-175, doi:10.1016/0006-8993(71)90358-1 (1971).
- 6 Eichenbaum, H. Time cells in the hippocampus: a new dimension for mapping memories. *Nature reviews. Neuroscience* **15**, 732-744, doi:10.1038/nrn3827 (2014).
- 7 Girardeau, G., Inema, I. & Buzsaki, G. Reactivations of emotional memory in the hippocampus-amygdala system during sleep. *Nature neuroscience* **20**, 1634-1642, doi:10.1038/nn.4637 (2017).

- 8 Hafting, T., Fyhn, M., Bonnevie, T., Moser, M. B. & Moser, E. I. Hippocampus-independent phase precession in entorhinal grid cells. *Nature* **453**, 1248-1252, doi:10.1038/nature06957 (2008).
- 9 Igarashi, K. M., Lu, L., Colgin, L. L., Moser, M. B. & Moser, E. I. Coordination of entorhinal-hippocampal ensemble activity during associative learning. *Nature* **510**, 143-147, doi:10.1038/nature13162 (2014).
- 10 Lundqvist, M. *et al.* Gamma and Beta Bursts Underlie Working Memory. *Neuron* **90**, 152-164, doi:10.1016/j.neuron.2016.02.028 (2016).
- 11 Siapas, A. G., Lubenov, E. V. & Wilson, M. A. Prefrontal phase locking to hippocampal theta oscillations. *Neuron* **46**, 141-151, doi:10.1016/j.neuron.2005.02.028 (2005).
- 12 Ito, H. T., Zhang, S. J., Witter, M. P., Moser, E. I. & Moser, M. B. A prefrontal-thalamo-hippocampal circuit for goal-directed spatial navigation. *Nature* **522**, 50-55, doi:10.1038/nature14396 (2015).
- 13 Sigala, N. & Logothetis, N. K. Visual categorization shapes feature selectivity in the primate temporal cortex. *Nature* **415**, 318-320, doi:10.1038/415318a (2002).
- 14 Walker, E. Y., Cotton, R. J., Ma, W. J. & Tolias, A. S. A neural basis of probabilistic computation in visual cortex. *Nature neuroscience* **23**, 122-129, doi:10.1038/s41593-019-0554-5 (2020).
- 15 Denfield, G. H., Ecker, A. S., Shinn, T. J., Bethge, M. & Tolias, A. S. Attentional fluctuations induce shared variability in macaque primary visual cortex. *Nat Commun* **9**, 2654, doi:10.1038/s41467-018-05123-6 (2018).
- 16 Ecker, A. S. *et al.* State dependence of noise correlations in macaque primary visual cortex. *Neuron* **82**, 235-248, doi:10.1016/j.neuron.2014.02.006 (2014).
- 17 Cohen, J. *Statistical power analysis for the behavioral sciences*. (Academic, 1977).

REVIEWERS' COMMENTS:

Reviewer #1 (Remarks to the Author):

I thank the authors for taking time to carefully address my concerns. The addition of data and analysis considerably strengthened this manuscript making it suitable for publication. I have three suggestions for the final version:

1. Consider adding your effect size calculations from the rebuttal to the actual manuscript, perhaps the supplemental material?
2. With regard to my original point 3) I apologise for not being more explicit. My concern was mainly whether the differential SI's could be caused by a bias for one or another area/layer in one animal vs the next, i.e. record mainly in EC LII vs EC LV or CA1 vs CA3, etc. The rebuttal addresses this by reporting no correlation between recording site and SI. I believe this information would help the reader interpret the findings and thus encourage the authors to include it in the manuscript/supplemental material.
3. The new lines 181 – 194 in the MS could do with some rephrasing to improve clarity and fix some mistakes (e.g. „did not different from“).

Reviewer #2 (Remarks to the Author):

The authors have addressed all concerns raised previously in the original review and have significantly improved the clarity of the manuscript. As stated in my original review, this is very well-written and rigorous study showing important data regarding the neural mechanisms of error driven learning in the MTL. I have no further comments.

REVIEWERS' COMMENTS:

Reviewer #1 (Remarks to the Author):

I thank the authors for taking time to carefully address my concerns. The addition of data and analysis considerably strengthened this manuscript making it suitable for publication.

The authors thank the reviewer's contribution in improving the manuscript and describe the modification per request bellow.

I have three suggestions for the final version:

1. Consider adding your effect size calculations from the rebuttal to the actual manuscript, perhaps the supplemental material?

All the effect size calculations were included in the manuscript as statistical reports. For example, in line 118-119 (paired t-test, $t(281) = 4.62, p < .001, d = 0.39$), the effect size (Cohen's d) is reported as d , adhering to APA style throughout the whole manuscript.

2. With regard to my original point 3) I apologize for not being more explicit. My concern was mainly whether the differential SI's could be caused by a bias for one or another area/layer in one animal vs the next, i.e. record mainly in EC LII vs EC LV or CA1 vs CA3, etc. The rebuttal addresses this by reporting no correlation between recording site and SI. I believe this information would help the reader interpret the findings and thus encourage the authors to include it in the manuscript/supplemental material.

The description of the correlation analysis is included in manuscript line 452-457.

3. The new lines 181 – 194 in the MS could do with some rephrasing to improve clarity and fix some mistakes (e.g. „did not different from“).

The lines were rephrased to increase clarity.

Reviewer #2 (Remarks to the Author):

The authors have addressed all concerns raised previously in the original review and have significantly improved the clarity of the manuscript. As stated in my original review, this is very well-written and rigorous study showing important data regarding the neural mechanisms of error driven learning in the MTL. I have no further comments.

The authors thank the reviewer's effort again and appreciate the kind remarks.